# Association Analysis of Maximum Standardized Uptake Values Based on ^18^F-FDG PET/CT and EGFR Mutation Status in Lung Adenocarcinoma

**DOI:** 10.3390/jpm13030396

**Published:** 2023-02-23

**Authors:** Jianxiong Gao, Yunmei Shi, Rong Niu, Xiaoliang Shao, Xiaonan Shao

**Affiliations:** 1Department of Nuclear Medicine, The Third Affiliated Hospital of Soochow University, Changzhou 213003, China; 2Changzhou Key Laboratory of Molecular Imaging, Changzhou 213003, China

**Keywords:** lung adenocarcinoma, positron emission tomography computed tomography, fluorodeoxyglucose F18, SUV_max_, EGFR

## Abstract

(1) Background: To investigate the association between maximum standardized uptake value (SUV_max_) based on ^18^F-FDG PET/CT and EGFR mutation status in lung adenocarcinoma. (2) Methods: A total of 366 patients were retrospectively collected and divided into the EGFR mutation group (*n* = 228) and EGFR wild-type group (*n* = 138) according to their EGFR mutation status. The two groups’ general information and PET/CT imaging parameters were compared. A hierarchical binary logistic regression model was used to assess the interaction effect on the relationship between SUV_max_ and EGFR mutation in different subgroups. Univariate and multivariate logistic regression was used to analyze the association between SUV_max_ and EGFR mutation. After adjusting for confounding factors, a generalized additive model and smooth curve fitting were applied to address possible non-linearities. (3) Results: Smoking status significantly affected the relationship between SUV_max_ and EGFR mutation (*p* for interaction = 0.012), with an interaction effect. After adjusting for age, gender, nodule type, bronchial sign, and CEA grouping, in the smoking subgroup, curve fitting results showed that the relationship between SUV_max_ and EGFR mutation was approximately linear (df = 1.000, c^2^ = 3.897, *p* = 0.048); with the increase in SUV_max_, the probability of EGFR mutation gradually decreased, and the OR value was 0.952 (95%CI: 0.908–0.999; *p* = 0.045). (4) Conclusions: Smoking status can affect the relationship between SUV_max_ and EGFR mutation status in lung adenocarcinoma, especially in the positive smoking history subgroup. Fully understanding the effect of smoking status will help to improve the accuracy of SUV_max_ in predicting EGFR mutations.

## 1. Introduction

Cancer is the leading cause of death in the world. Non-small cell lung cancer (NSCLC) accounts for 80–85% of all lung cancer cases [1], and its mortality and morbidity are the highest in China [2]. EGFR mutations play a crucial role in tumor aggressiveness and response to therapy in NSCLC. Tumors with EGFR mutations tend to have higher potential for proliferation, invasion, angiogenesis, and metastasis compared to EGFR wild-type tumors [3]. However, EGFR mutations also make these tumors sensitive to EGFR tyrosine kinase inhibitors (TKIs) [4]. Lung adenocarcinoma is the most common histological type of non-small cell lung cancer [1] and is more prone to EGFR mutations than other subtypes [5]. Studies have shown that EGFR TKIs effectively prolong the progression-free survival (PFS) of lung adenocarcinoma patients with EGFR mutations [6,7]. Thus, identifying EGFR mutation status can help improve patient outcomes.

The commonly used pathology technique to assess EGFR mutation status is the biopsy, which is time-consuming and expensive, and its accuracy is affected by tumor heterogeneity [8,9]; moreover, many patients cannot be tested through biopsy due to health conditions and other reasons. Sometimes, blood samples are used instead of biopsies for testing EGFR mutations, but a meta-analysis [10] found that the pooled sensitivity (65%) of detecting EGFR mutation from blood samples was low. Therefore, it is necessary to develop non-invasive imaging methods, such as ^18^F-FDG PET/CT, to predict EGFR mutation status.

Several semiquantitative metabolic parameters, such as SUV_max_, metabolic tumor volume (MTV), and total lesion glycolysis (TLG), which reflect ^18^F-FDG metabolic activity as measured by PET/CT, have been shown to be effective in predicting the status of EGFR mutation in patients with NSCLC [11]. However, some studies have found that MTV and TLG are not independent predictors of EGFR mutation status in NSCLC patients, whereas SUV_max_ is [12,13,14]. Therefore, SUV_max_ may be a more reliable and convenient parameter for assessing the relationship between EGFR mutation status and glucose metabolism in NSCLC. The maximum standard uptake value (SUV_max_) is a simple and widely used parameter of PET/CT, reflecting the highest uptake value of ^18^F-FDG in tumor tissue and representing the metabolic state of glycolysis in cancer cells. Previous studies that explored the relationship between SUV_max_ and EGFR mutations did not reach consistent conclusions [15,16,17,18], which might be due to: (1) The heterogeneity of the study population, such as different TNM stages and histological types of the enrolled patients; in addition, most studies did not distinguish between solid and subsolid lesions. (2) The sample size of some studies was less than 200 cases [17], and the results may be biased.

To overcome the above problems, we analyzed the association between SUV_max_ and EGFR mutation in lung adenocarcinoma patients to clarify the relationship between PET/CT metabolic parameters and EGFR mutation and assist lung function adenocarcinoma targeted therapy.

## 2. Materials and Methods

### 2.1. Patient Selection and Characteristics

We retrospectively analyzed lung adenocarcinoma patients who underwent ^18^F-FDG PET/CT examination in the Third Affiliated Hospital of Soochow University from January 2018 to December 2020. Inclusion criteria: (1) lung adenocarcinoma was confirmed by surgery or biopsy pathology, and the pathological classification was based on IASLC/ATS/ERS lung adenocarcinoma classification criteria [19]; (2) the patients completed ^18^F-FDG PET/CT examination before surgery, and the interval between surgery and examination was less than 30 days; (3) the EGFR test result was clear; (4) patient had no history of other malignant tumors. Exclusion criteria: (1) lesions with poor image quality or were difficult to measure; (2) lesions were other pathological subtypes; (3) patient had no chest thin-slice CT; (4) patient had severe liver disease or diabetes.

Patient information, including age, gender, smoking status, clinical stage, tumor markers, imaging features, ^18^F-FDG PET/CT semi-quantitative analysis results, and postoperative pathology, were recorded. This study followed the Helsinki Declaration of Principles and was approved by the Ethics Committee of our institution No. [2022] KD 087.

A total of 366 patients with lung adenocarcinoma were finally enrolled in this study, including 194 females and 172 males, with an average age of 64.1 ± 9.2 years (from 38 to 85 years old). In total, 131 (35.8%) patients had a positive smoking history, and 130 (35.5%) patients had subsolid nodules. There were 175 cases (47.8%) in clinical stage I, 14 cases (3.8%) in stage II, 59 cases (16.1%) in stage III, and 118 cases (32.2%) in stage IV. The pathology was confirmed by surgery, puncture, or bronchoscopy. There were 138 cases (37.7%) of EGFR wild-type and 228 cases (62.3%) of EGFR mutation (2 cases on exon 18, 96 cases on exon 19, 9 cases on exon 20, 114 cases on exon 21, 1 case on exon 19 and 20, 2 cases on exon 19 and 21, and 4 cases unknown). The study flow chart is shown in Figure 1.

### 2.2. PET/CT Imaging

The German Siemens BiographmCT (64) PET/CT machine was used for PET/CT image acquisition. The patients fasted for 4–6 h before examination, and their height, weight, and blood sugar were recorded on the day of examination. ^18^F-FDG was injected intravenously according to the patient’s body weight at 3.70–5.55 MBq/kg, and the imaging agent was purchased from Nanjing Jiangyuan Andike Positron Research and Development company (radiochemical purity > 95%). Patients rested in a quiet and comfortable environment for 1 h before PET/CT whole-body imaging. The patient was placed supine during scanning and kept holding the head with both hands. The scanning time was 2 min/bed. The collection range was from the base of the skull to the middle of the femur. Low-dose CT imaging was performed before the PET scan. After image acquisition, the TrueX + TOF (ultraHD-PET) system was used for image reconstruction and interpretation. A post-processing workstation TrueD system (Siemens) was used for image evaluation.

Scanning and reconstruction parameters: tube voltage 140 kV, tube current was automatically adjusted by caredose software according to human anatomy and tissue density, rotation time 0.5 s/turn, pitch 0.6, slice thickness 3.0 mm, matrix 512 × 512, lung window (window width 1200 HU, window level −600 HU), mediastinal window (window width 350 HU, window level 40 HU). Image reconstruction was performed based on a slice thickness of 1.0 mm and slice interval of 0.5 mm.

### 2.3. Image Analysis

Two nuclear medicine doctors, each with over ten years of experience, recorded and examined all images without knowledge of the patients’ EGFR test results. In case of disagreement, the two doctors reached an agreement after discussion. On thin-slice CT, the lesion type (solid, subsolid), location (upper, middle, and lower lobe of right lung and upper and lower lobe of left lung), shape (circular/oval, polygonal/irregular), the presence or absence of cystic components in the lesion, lobulation sign, burr sign, bronchial sign, vacuole sign, pleural indentation sign, long diameter and short diameter of the lesion (measured at the largest cross-section of the lesion, the long and short diameter measurement lines were perpendicular) were recorded. The PET image parameter we used was the SUV_max_ of the lesion. The measurement of SUV_max_ was performed on PET-CT fusion images. The aspherical volume of interest (VOI) that contained the lesion was selected for SUV_max_ measurement. The measurements from 2 physicians were averaged and recorded.

### 2.4. EGFR Mutation Test

The mutations on EGFR exons 18, 19, 20, and 21 were tested on surgical specimens or biopsy tissues using an allele-specific amplification method. The Shanghai Yuanqi EGFR gene mutation detection kit was used for EGFR gene mutation detection, and the results were determined according to the manuals provided by the kit.

### 2.5. Statistical Analysis

Statistical analysis was performed using R software (version 3.4.3; http://www.R-project.org/, accessed on 21 March 2022). Continuous variables were expressed as Mean (standard deviation) (normal distribution) or Median (Q1–Q3) (skewed distribution); categorical variables were expressed as frequency or percentage (%). An χ^2^ test (categorical variable), *t*-test (normal distribution), or Mann–Whitney U test (skewed distribution) was used to compare the differences in general data and imaging characteristics between different EGFR mutation statuses (binary variables).

After adjusting for age, we used a hierarchical binary logistic regression model to assess whether there was an interaction effect on the relationship between SUV_max_ and EGFR mutations in different subgroups. The effect size with a 95% confidence interval was recorded.

In the positive smoking history subgroup, we used univariable and multivariable logistic regression methods to examine the association between SUV_max_ and EGFR mutation to construct three different models, including unadjusted, preliminarily adjusted, and fully adjusted models. In multivariable regression analysis, when a factor was introduced into the basic model or excluded from the complete model if the regression coefficient of SUV_max_ changed by more than 10% or the factor was significantly associated with EGFR mutation (*p* < 0.1), then it was included in the final model as a potential confounding factor. To test the robustness of the results, we performed a sensitivity analysis, transforming SUV_max_ into categorical variables by tripartition and calculating *p* values for trend. We also used generalized additive models and smooth curve fitting to account for possible non-linearity. All statistical tests were two-sided, and *p* < 0.05 was considered statistically significant. Forty-four (12%) patients had missing CEA data. No data imputation was used for missing CEA data. We included patients with CEA missing as a separate subgroup in the interaction analysis.

## 3. Results

### 3.1. The General Data, Morphological Features, SUV_max_, and Pathological Subtypes of the Two Patient Groups

We found that the proportions of males, positive smoking histories, and solid nodules in the wild-type group were significantly higher than those in the mutant group (68.1% vs. 34.2%, 56.5% vs. 23.3%, 76.1% vs. 57.5%, respectively; *p* < 0.001). The proportions of bronchial sign, pleural indentation sign, and vascular bundle sign in the mutant group were significantly higher than those in the wild-type group (61.4% vs. 42.8%, 72.8% vs. 51.5%, 64.0% vs. 50.7%, respectively; *p* < 0.05). The wild-type group’s tumor long axes, short axes, and clinical stages were significantly higher than those in the mutant group (all *p* < 0.01). For tumor indicators, the level of CEA in the wild-type group was higher than that in the mutant group, but the difference was not significant (*p* = 0.096). The SUV_max_ in the wild-type group was significantly higher than that in the mutant group (*p* = 0.003) (Table 1).

### 3.2. Interaction Analysis

After adjusting for age, we compared the relationship between SUV_max_ and EGFR mutations in different subgroups, including gender, smoking history, nodule type, shape, lobulation, burr sign, bronchial sign, vacuolar sign, pleural indentation sign, vascular bundle sign, tumor long axis (three groups), clinical stage (two groups), and CEA (three groups) (Figure 2). The results showed that smoking history and CEA grouping significantly affected the relationship between SUV_max_ and EGFR mutation in lung adenocarcinoma (both *p* < 0.05), suggesting that there was an interaction effect (Figure 2). Further analysis found that in the smoking subgroup, the OR value of SUV_max_ and EGFR mutation was 0.947 (95%CI: 0.909–0.986; *p* = 0.008). In the normal CEA subgroup, the OR value of SUV_max_ and EGFR mutation was 0.939 (95%CI: 0.906–0.973; *p* < 0.001). The characteristics of different smoking history groups were compared in Appendix A.

### 3.3. Multivariable Regression for the Association between SUV_max_ and EGFR Mutation Probability in Smoking Subgroups

Table 2 shows the univariable and multivariable logistic regression analyses for continuous SUV_max_ and tripartite SUV_max_ [44 cases in Tertile 1 (SUV_max_: 0.94–9.98), 43 cases in Tertile 2 (SUV_max_: 10.61–17.48), and 44 cases in Tertile 3 (SUV_max_: 17.57–67.17)]. Unadjusted covariates were equivalent to univariable logistic regression analysis. Preliminarily adjusted covariates included age and gender. Fully adjusted covariates included age, gender, nodule type, bronchial sign, and CEA grouping. For continuous SUV_max_, the increase in SUV_max_ was associated with decreased probability of EGFR mutation in unadjusted, preliminarily adjusted, and fully adjusted regression equations, and the ORs were 0.948, 0.946, and 0.952, respectively (*p* < 0.05 for all).

For tripartite SUV_max_, the increasing trend of SUV_max_ was significantly associated with decreased probability of EGFR mutation in the preliminarily adjusted regression equations (*p* for trend = 0.039). Nevertheless, the trends mentioned above are not obvious in unadjusted and fully adjusted regression equations (*p* for trend >0.05 for all), only the trend for Tertile 2 vs. Tertile 1 was significant (ORs were 0.294 and 0.292, *p* < 0.05 for both).

### 3.4. Curve Fitting

In the analysis of smoking subgroups, the GAM test results showed an approximately linear relationship between SUV_max_ and EGFR mutation after adjusting for age, gender, nodule type, bronchial sign, and CEA grouping (df = 1.000, c^2^ = 3.897, *p* = 0.048). With the increase in SUV_max_, the probability of EGFR mutation significantly reduced, and the OR value was 0.952 (95%CI: 0.908–0.999; *p* = 0.045) (Figure 3A).

When grouping patients based on SUV_max_ tertiles, the relationship between different tertiles of SUV_max_ and EGFR mutation probability had a similar trend in the smoking group [56.8% (25/44), 27.9% (12/43), and 36.4% (16/44), *p* = 0.018]. After adjusting for age, gender, nodule type, bronchial sign, and CEA grouping, the probabilities of EGFR mutation with Tertile 1 to Tertile 3 were 56.8% (95%CI: 35.7–75.7%), 28.3% (95%CI: 14.9–47.1%), and 35.6% (95%CI: 19.4–56.0%) (Figure 3B).

## 4. Discussion

This study investigated the association between PET/CT-derived SUV_max_ and EGFR mutation status in 366 patients with lung adenocarcinoma. Through interaction analysis, we found that smoking status affected the relationship between SUV_max_ and EGFR mutation, especially in patients with a positive smoking history. After fully adjusting covariates in smoking patients with the increase in SUV_max_, we found the probability of EGFR mutation gradually decreased, and there was an approximately linear relationship between the two.

Consistent with the previous epidemiological studies [6], more males and patients with positive smoking histories were in the EGFR wild-type group. Moreover, the proportion of subsolid nodules in the mutant group was higher, suggesting that the presence of ground-glass components was associated with EGFR mutation, which was consistent with previous studies [20,21]. At the same time, the size and clinical stage of the EGFR wild-type group were significantly higher than those of the mutant group, suggesting that the lesions in the wild-type group were larger and most advanced. According to the IASLC grading system proposed by the Pathology Committee of the International Association for the Study of Lung Cancer in 2020 [22], Fujikawa et al. [23] found that specific features of the lesion (such as solid nodules, male, and a positive smoking history) could predict the lesion differentiation stage as III. The probability of EGFR mutation was low and the prognosis was poor, which is also consistent with our findings. In addition, the EGFR mutation group was more prone to bronchial sign, vascular bundle sign, and pleural indentation sign, which might be due to the fact that the nodules in the mutation group were smaller and showed more diverse features.

Unlike previous findings, Gu et al. [24] demonstrated that a higher carcinoembryonic antigen (CEA) level was a significant predictor of EGFR mutation. However, we found that the level of CEA in the EGFR wild-type group was higher than that of the EGFR mutation group, but the difference was not significant. The discrepancy might be due to different study populations. In our study, the wild-type group had more advanced-stage patients, which exhibited greater tumor burden and higher CEA levels.

^18^F-FDG is a glucose analog. The glucose transporter-1 (GLUT-1) plays an important role in tumor ^18^F-FDG uptake, and inhibition of GLUT-1 expression can significantly reduce the tumor uptake of ^18^F-FDG [25]. Through cellular and molecular experiments, Chen et al. [26] found that EGFR mutations reduced the uptake of FDG in NSCLC via the NOX4/ROS/GLUT1 pathway, indicating that the uptake of ^18^F-FDG is related to EGFR mutation status and that it is possible to use FDG PET/CT metabolic parameters to predict tumor EGFR mutation status. Previous studies have yielded conflicting results regarding the relationship between SUV_max_ and EGFR mutation status. Gu et al. [24] and Cho et al. [27] demonstrated that low SUV_max_ was a significant predictor for EGFR mutations, which is consistent with our findings that SUV_max_ was negatively correlated with EGFR mutation probability. However, Liu et al. [28] and Caicedo et al. [15] found no significant correlation between EGFR mutations and SUV_max_. Ko et al. [29] found that high SUV_max_ was significantly associated with EGFR mutations in lung adenocarcinoma. The study by Caicedo et al. [15] only included patients with stages III-IV NSCLC, and the study by Liu et al. [28] did not include stage I patients, while our study included patients with all clinical stages, and the proportion of stage I patients was high. The study by Ko et al. [29] excluded tumors with a maximum diameter of <1 cm, but it has been demonstrated [30,31] that tumor diameter is also a relevant factor for EGFR mutations. Moreover, the studies by Caicedo et al. [15] and Ko et al. [29] did not consider the influence of nodule type and smoking history, which may cause inconsistency with our study.

Through further stratified analysis, we found smoking history and CEA had an interaction effect on the relationship between SUV_max_ and EGFR mutation status. In patients with a positive smoking history, the proportions of males, large lesions, and solid nodules were higher, and previous studies [6,23,32] found that these characteristics were associated with lower EGFR mutation rates. In the negative smoking history subgroup, female and small lesions were the predominant features, and the proportion of solid nodules and subsolid nodules was relatively balanced. The different characteristics of smoking history subgroups may account for the interaction effect. The normal CEA subgroup had a large sample size (*n* = 180) in different CEA subgroups, which may account for the more significant relationship between SUV_max_ and EGFR.

In the subgroup with a positive smoking history, we found that the probability of EGFR mutation decreased linearly with the increase in SUV_max_. This approximate linear relationship suggests that in subsequent studies, in addition to the influence of smoking history, it may be better to use a linear model when establishing the prediction model for EGFR mutation status, such as a multivariate logistic regression model [33,34]. Considering that the samples with high SUV_max_ were too few, we further divided the SUV_max_ into three equal parts. With the increase in the SUV_max_ levels, the probability of EGFR mutation still had a similar trend, indicating that the relationship between the two was robust.

### Strengths and Limitations of This Study

Advantages: 1. This study was a single-center study with a large sample size, which avoided the confounding imaging results from different centers. 2. The patients covered all stages of lung adenocarcinoma (I–IV), and the effect of nodule type was considered; therefore, the patient population was representative. 3. Our study fully considered various influencing factors; we not only adjusted confounding factors but also performed stratified analysis and curve fitting. Limitations: 1. This study is a retrospective study, the subject selection may be biased, and the results can only represent the Chinese population. 2. All patients had lung adenocarcinoma, and the results cannot be extended to other lung cancer subtypes. 3. The sample size of some subgroups was still small, such as the stage II subgroup; thus, the sample size needs to be further expanded. 4. Compared to SUV_max_, radiomics features can better reflect the spatial distribution of tumors and more comprehensively evaluate tumor heterogeneity. In recent years, using machine learning methods to evaluate radiomic features and predict EGFR mutation status has become a research “hot spot” [30,33,35,36]. In a follow-up work, we will conduct research related to radiomics and machine learning.

## 5. Conclusions

After adjusting for confounding factors, we found that smoking status significantly affected the relationship between SUV_max_ and EGFR mutation status. There was an approximately linear relationship between SUV_max_ and EGFR mutation status in the subgroup with a positive smoking history. With the increase in SUV_max_ and the stepwise elevation of SUV_max_ levels, the probability of EGFR mutation gradually decreased. Fully understanding the influence of smoking status on the predictive value of SUV_max_ can help improve its accuracy in predicting EGFR mutations and develop more specific models. Our results demonstrate that ^18^F-FDG PET/CT imaging can be an effective non-invasive approach to predict EGFR mutations in lung adenocarcinoma patients.

## Figures and Tables

**Figure 1 jpm-13-00396-f001:**
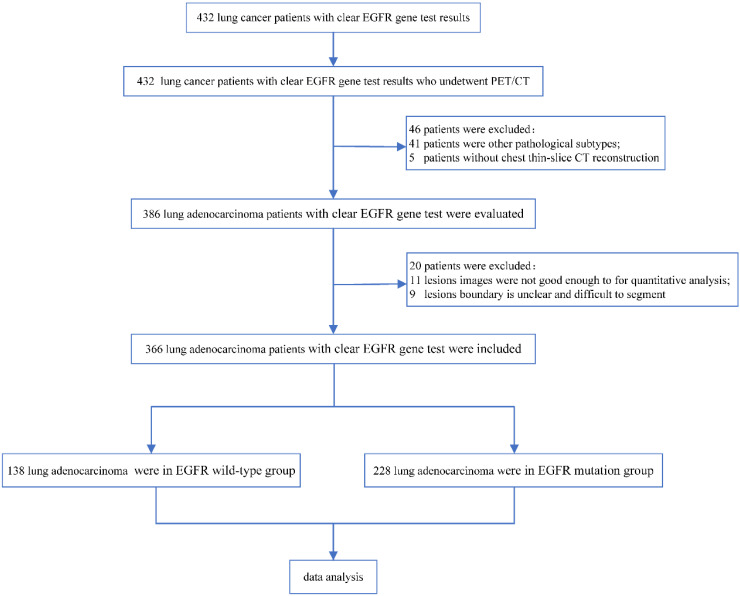
Flowchart of patient selection. EGFR, epidermal growth factor receptor.

**Figure 2 jpm-13-00396-f002:**
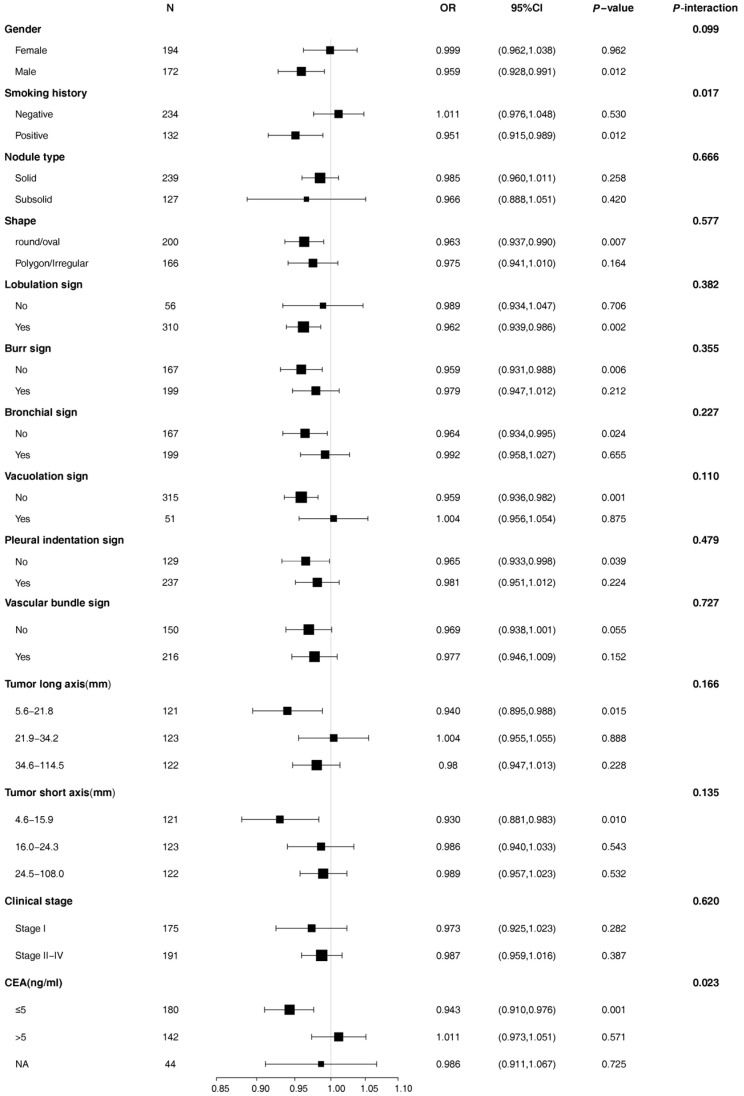
The stratification analysis of the association between SUV_max_ and the probability of EGFR mutation (adjusted for age).

**Figure 3 jpm-13-00396-f003:**
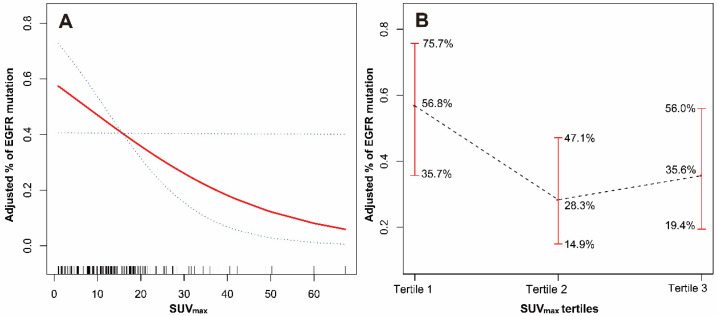
(**A**) The relationship between SUV_max_ and the probability of EGFR mutation (the solid red line indicates the fitted line of the probability of EGFR mutation and SUV_max_; the blue dotted line is the 95% confidence interval; adjusted for age and CEA grouping). (**B**) The relationship between SUV_max_ tertiles and the probability of EGFR mutation (the black dotted line indicates the fitted line of the probability of EGFR mutation and SUV_max_ tertiles; the red line is the 95% confidence interval; adjusted for age and CEA grouping).

**Table 1 jpm-13-00396-t001:** Comparison of general data and imaging parameters between EGFR mutant and wild-type groups.

EGFR	Wild-Type	Mutant	*p*-Value
*n*	138	228	
Age (years)	64.9 (9.2)	63.6 (9.2)	0.209
Gender			<0.001
Female	44 (31.9%)	150 (65.8%)	
Male	94 (68.1%)	78 (34.2%)	
Smoking history	78 (56.5%)	53 (23.3%)	<0.001
Nodule type			<0.001
Solid	105 (76.1%)	131 (57.5%)	
Subsolid	33 (23.9%)	97 (42.5%)	
Location			0.807
Upper right	42 (30.4%)	77 (33.8%)	
Middle right	6 (4.4%)	14 (6.1%)	
Lower right	27 (19.6%)	45 (19.7%)	
Upper left	38 (27.5%)	59 (25.9)	
Lower left	25 (18.1%)	33 (14.5%)	
Shape			0.153
round/oval	82 (59.4%)	118 (51.8%)	
Polygon/Irregular	56 (40.6%)	110 (48.3%)	
Lobulation sign	116 (84.1%)	194 (85.1%)	0.791
Burr sign	71 (51.5%)	128 (56.1%)	0.383
Bronchial sign	59 (42.8%)	140 (61.4%)	<0.001
Vacuolation sign	21 (15.2%)	30 (13.2%)	0.581
Pleural indentation sign	71 (51.5%)	166 (72.8%)	<0.001
Vascular bundle sign	70 (50.7%)	146 (64.0%)	0.012
Tumor long axis (mm)	32.0 (20.6–45.3)	25.0 (19.8–35.2)	0.001
CEA (ng/mL)	4.71 (2.5–14.6)	3.5 (1.6–12.3)	0.096
Clinical stage			0.004
I	49 (35.5%)	126 (55.3%)	
II	11 (8.0%)	3 (1.3%)	
III	27 (19.6%)	32 (14.0%)	
IV	51 (37.0%)	67 (29.4%)	
SUV_max_	12.3 (5.8–17.5)	9.0 (3.3–16.5)	0.004

Note: Mean (SD)/Median (Q1–Q3)/*n* (%).

**Table 2 jpm-13-00396-t002:** Multivariable regression for the association between SUV_max_ and EGFR mutation probability in smoking subgroups.

Exposure	Non-Adjusted	Adjust I	Adjust II
	OR (95%CI) *p* Value	OR (95%CI) *p* Value	OR (95%CI) *p* Value
SUV_max_	0.948 (0.912, 0.986) 0.007	0.946 (0.907, 0.986) 0.009	0.952 (0.908, 0.999) 0.045
SUV_max_ Tertile			
Tertile 1 (0.94–9.98) *n* = 44	1.0	1.0	1.0
Tertile 2 (10.61–17.48) *n* = 43	0.294 (0.120, 0.720) 0.007	0.263 (0.103, 0.673) 0.005	0.292 (0.105, 0.811) 0.018
Tertile 3 (17.57–67.17) *n* = 44	0.434 (0.184, 1.022) 0.056	0.387 (0.156, 0.962) 0.041	0.393 (0.136, 1.134) 0.084
*p* for trend	0.052	0.039	0.090

Non-adjusted model, adjust for: none. Adjust I model, adjust for: age; gender. Adjust II model, adjust for: age; gender; nodule type; bronchial sign; CEA grouping.

## Data Availability

The data are not publicly available due to privacy reasons.

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
