# Peer review of "Association Analysis of Maximum Standardized Uptake Values Based on 18F-FDG PET/CT and EGFR Mutation Status in Lung Adenocarcinoma"

_jpm, 2023, doi:10.3390/jpm13030396_

Round 1

Reviewer 1 Report

General comments:

This is a well written paper, with thorough statistical analysis on a large cohort of lung cancer patients looking at correlations between EGFR and PET-based SUVmax as a function of smoking history. Below are my comments to further improve the manuscript:

1.     Introduction: The authors should add a paragraph on EGFR in lung cancer and its role in tumour aggressiveness and response to therapy. The existing information on EGFR in the paper is too scarce.

2.     The justification of the study is not very clear. What is the biological basis behind electing to find a correlation between EGFR and SUVmax; why SUVmax and not other metabolic parameter, such as metabolic tumour volume or total lesion glycolysis?

3.     Results – the first paragraph is not a result. This belongs to Methods. Please move the data to section 2.1 and rename the title to ‘Patients selection and characteristics’

Specific comments:

1.     Abstract, line 22 – A total of 366 patients were…

2.     Section 2.3, lines 110-111 – first sentence needs rewriting

3.     Line 118 – ‘The PET image parameter was the SUV max of the lesion’ – this sentence requires rewording (see my comments above regarding SUV); say that the chosen parameter for the purpose of this study was SUVmax (based on the justification that the authors should add to the Intro).

4.     Line 121 – ‘The measurements from 2 physicians were averaged and recorded.’ – irrespective of the differences?

Author Response

Point 1: Introduction: The authors should add a paragraph on EGFR in lung cancer and its role in tumour aggressiveness and response to therapy. The existing information on EGFR in the paper is too scarce.

Response 1: Thanks for your comments. I agree with your suggestion, and I have added a paragraph on EGFR in lung cancer, its role in tumor aggressiveness, and its response to therapy in the Introduction section . (see Introduction Section, line46-55)

Point 2: The justification of the study is not very clear. What is the biological basis behind electing to find a correlation between EGFR and SUVmax; why SUVmax and not other metabolic parameter, such as metabolic tumour volume or total lesion glycolysis?

Response 2:  Thanks for your comments. In Discussion section , we described the biological basis behind electing to find a correlation between EGFR and SUVmax .(see Discussion section, line316-321)

The rationale for selecting SUVmax over other metabolic parameters has been included in the third paragraph of the Introduction section. (see Introduction section, line64-70)

Point 3: Results – the first paragraph is not a result. This belongs to Methods. Please move the data to section 2.1 and rename the title to ‘Patients selection and characteristics’.

Response 3: Thanks for your comments. The first paragraph of the Results section has been moved to the Methods section, and the title of the section has been changed to "Patient Selection and Characteristics."(see section 2.1, line110-118)

Point 4: Abstract, line 22 – A total of 366 patients were…

Response 4: The above-mentioned issue has been duly revised. We appreciate your efforts in providing the necessary corrections.

Point 5: Section 2.3, lines 110-111 – first sentence needs rewriting.

Response 5: Thanks.We have rewritten the first sentence of Section 2.3, lines 110-111.

Point 6: Line 121 – ‘The measurements from 2 physicians were averaged and recorded.’ – irrespective of the differences?

Response 6: Thanks for your comments. For relatively large lesions, the SUVmax measured by both doctors should be the same. This is because our SUVmax is determined from the maximum SUV value point in the VOI, and the average value obtained by the two doctors is primarily to minimize differences in VOI delineation caused by small lesions.

Reviewer 2 Report

The article entitled  “Association analysis of maximum standardized uptake values based on 18F-FDG PET/CT and EGFR mutation status in lung adenocarcinoma” investigated retrospectively 366 cases with lung adenocarcinoma and with evidence of EGFR gene in order to analyze the association between maximum standardized uptake value (SUVmax) based on 18F-FDG PET/CT and EGFR mutation status in lung adenocarcinoma.

I retain the article very interesting and well developed although it is necessary in my opinion to include in the discussion some references concerning the  artificial intelligence and radiomics which are assuming more and more importance in oncological diagnostics in general and in lung adenocarcinoma  in particular.

Some articles evaluated the Form Factors in CT scan as Potential Imaging Biomarkers to Differentiate Benign and Malignant Lung Lesions and some authors applied the evaluation of shape and Texture Features from 18F-FDG PET/CT in order to discriminate between Benign and Malignant Solitary Pulmonary Nodules.

Moreover, recently a comparative evaluation of conventional and deep learning methods for semi-automated segmentation of pulmonary nodules on CT was also considered.

Considering the recent scientific aspects emerging from the literature, the discussion will acquire greater depth and a broader vision.

Author Response

Point : The article entitled  “Association analysis of maximum standardized uptake values based on 18F-FDG PET/CT and EGFR mutation status in lung adenocarcinoma” investigated retrospectively 366 cases with lung adenocarcinoma and with evidence of EGFR gene in order to analyze the association between maximum standardized uptake value (SUVmax) based on 18F-FDG PET/CT and EGFR mutation status in lung adenocarcinoma.

I retain the article very interesting and well developed although it is necessary in my opinion to include in the discussion some references concerning the  artificial intelligence and radiomics which are assuming more and more importance in oncological diagnostics in general and in lung adenocarcinoma  in particular.

Some articles evaluated the Form Factors in CT scan as Potential Imaging Biomarkers to Differentiate Benign and Malignant Lung Lesions and some authors applied the evaluation of shape and Texture Features from 18F-FDG PET/CT in order to discriminate between Benign and Malignant Solitary Pulmonary Nodules.

Moreover, recently a comparative evaluation of conventional and deep learning methods for semi-automated segmentation of pulmonary nodules on CT was also considered.

Considering the recent scientific aspects emerging from the literature, the discussion will acquire greater depth and a broader vision.

Response : Thanks for your comments. We have added some content regarding radiomics and machine learning to the Discussion section. (see Discussion section, line368-372)
